# Increase in Serum MMP-9 and TIMP-1 Concentrations during Alcohol Intoxication in Adolescents—A Preliminary Study

**DOI:** 10.3390/biom12050710

**Published:** 2022-05-16

**Authors:** Katarzyna Zdanowicz, Monika Kowalczuk-Kryston, Witold Olanski, Irena Werpachowska, Wlodzimierz Mielech, Dariusz Marek Lebensztejn

**Affiliations:** 1Department of Pediatrics, Gastroenterology, Hepatology, Nutrition and Allergology, Medical University of Bialystok, ul. Waszyngtona 17, 15-274 Bialystok, Poland; monika.kowalczuk@udsk.pl (M.K.-K.); irena.werpachowska@udsk.pl (I.W.); lebensztejn@hoga.pl (D.M.L.); 2Department of Pediatric Emergency Medicine, Medical University of Bialystok, ul. Waszyngtona 17, 15-274 Bialystok, Poland; witold.olanski@udsk.pl (W.O.); wlodzimierz.mielech@udsk.pl (W.M.)

**Keywords:** liver fibrosis, MMP-9, TIMP-1, ethanol, adolescents

## Abstract

Background: Alcohol consumption by adolescents is responsible for a number of adverse health and social outcomes. Despite the well-established effect of alcohol use on the development of alcoholic liver disease, the relationship between the pattern of alcohol consumption and liver fibrosis is still unclear. This study is a follow-up to work on liver damage from alcohol intoxication. The aim of our study was to explore the early effects of alcohol intoxication on liver fibrosis in adolescents. Methods: The prospective study included 57 adolescents aged 14–17 years admitted to the emergency department (ED) from February 2017 to June 2018 due to acute alcohol intoxication. Serum levels of amino terminal propeptide of type III procollagen (PIIINP), type IV collagen, matrix metallopeptidase 9 (MMP-9) and tissue inhibitor of metalloproteinase 1 (TIMP-1) were determined by enzyme-linked immunosorbent assays. Results: There were significant differences in MMP-9 (*p* = 0.02) and TIMP-1 (*p* = 0.007) levels between the study and control groups. Liver parameters and selected markers of fibrosis were similar in groups in terms of blood alcohol concentrations (BAC). MMP-9 was positively correlated with alanine aminotransferase (ALT) (r = 0.38; *p* = 0.004) and total bilirubin (r = 0.39; *p* = 0.004). Positive significant correlations were also found between TIMP-1 and ALT (r = 0.47; *p* < 0.001), AST (r = 0.29; *p* = 0.03) and total bilirubin (r = 0.32; *p* = 0.02). In receiver operating characteristic (ROC) analysis, MMP-9 (AUC = 0.67, *p* = 0.02) and TIMP-1 (AUC = 0.69, *p* = 0.003) allowed for the differentiation of patients with and without alcohol intoxication. Conclusion: Our results show that even a single episode of alcohol intoxication in adolescents can lead to imbalance in markers of fibrosis.

## 1. Introduction

In 2018, the World Health Organization (WHO) estimated that alcohol consumption is responsible for more than 200 diseases and injury conditions [1]. Alcohol consumption is a well-known modifiable risk factor for the development of several diseases, including in the heart, brain, gastrointestinal tract and liver [2].

Ethanol is absorbed in the stomach and small intestine by diffusion and then is transferred to the liver via the portal blood stream [3]. The liver, as the first organ of alcohol metabolism, is particularly exposed to damage. Alcohol-associated liver disease (ALD) is a wide spectrum condition involving hepatic steatosis, alcoholic hepatitis (AH), alcohol-associated cirrhosis (AC) and acute AH presenting as acute-on chronic liver failure [4]. Hepatic metabolism of the ethanol generates substances damaging the liver by triggering inflammation, extracellular matrix (ECM) remodeling, fibrogenesis and fibrolysis [5]. Liver fibrosis is caused by disturbance in the accumulation of ECM components and its resorption, and leads to the formation of a fibrous scar [6]. It is a dynamic process of progression and regression, and the components of ECM are released into circulation [7]. Non-invasive serum markers of liver function indicating the process of organ fibrosis are, among others, amino terminal propeptide of type III procollagen (PIIINP), type IV collagen, matrix metallopeptidase 9 (MMP-9) and the tissue inhibitor of metalloproteinase 1 (TIMP-1). TIMP-1 interacts with the active form of MMP-9 in a 1:1 ratio [8]. Disturbances in the balance between them may indicate the degradation of ECM proteins and tissue remodeling [9]. Well-known markers of fibrosis are: PIIINP and TIPM-1, and markers of fibrinolysis are MMPs [10]. PIIINP is released during the synthesis and deposition of type III collagen. The number of molecules produced and PIIINP released is in a stoichiometric ratio of 1:1, so this protein is useful in the evaluation of fibrogenesis processes [11]. MMPs are calcium-dependent zinc-containing peptidases. They are useful markers of fibrinolysis due to their participation in the degradation and turnover of most components in the ECM. TIMPs, as inhibitors of MMPs activity, take part in the process of fibrosis [12].

Toxic and metabolic effects of alcohol are determined by the total volume of alcohol consumed and the pattern of drinking [1]. According to the “Recommendations for Alcohol Intake” from the 2015–2020 Dietary Guidelines for Americans, alcohol should only be consumed by adults and limited to no more than one drink per day for women and two drinks per day for men [13]. However, in the latest study conducted in adult healthy individuals and patients with chronic liver disease, a standardized, single dose of alcohol may cause early, excessive liver fibrogenesis [14]. In our prior study, we demonstrated serum total keratin-18 (M65) as a non-invasive marker of hepatocyte injury and apoptosis during alcohol intoxication in adolescents [15]. For this reason, we continue assessing the effects of alcohol consumption on liver injury in pediatric patients. The aim of our current study was to explore the outcome of alcohol intoxication on the biomarkers of liver fibrosis. To the best of our knowledge, the effect of a single episode of alcohol use on the liver’s fibrogenic response in adolescents has not been evaluated to date.

## 2. Materials and Methods

The design of the primary study is reported elsewhere [15] and is summarized briefly in this paper. The study included 57 adolescents aged 14–17 years admitted to the emergency department (ED) from February 2017 to June 2018 due to acute alcohol intoxication. The control group consisted of 24 children with confirmed lack of any liver diseases (based on the patient’s medical history, physical examination and laboratory results). Inclusion and exclusion criteria remain unchanged as previous study.

All participants underwent a physical examination. The sera were drawn from patients and healthy controls by venipuncture. Routine biochemical analyses including alanine aminotransferase (ALT), and additionally in study group total bilirubin, creatine kinase (CK), creatine kinase-MB (CK-MB) and urea, were measured on the day of blood collection by standard clinical laboratory techniques, using an automatic analyzer Dimension AR (Dade Behring, Newark, NJ, USA). Blood alcohol concentration (BAC) determinations were performed with headspace gas chromatography according to Cordell et al. [16].

The serum samples were collected after centrifugation at 4 °C for 15 min at 1000× *g* and immediately frozen at −80 °C until determinations of PIIINP, collagen Type IV, MMP-9 and TIMP-1. Serum PIIINP, collagen Type IV, MMP-9 and TIMP-1 levels were measured using a double-antibody sandwich ELISA with laboratory kits (Cloud-Clone Corp., Katy, TX, USA). Collected samples from all participants were analyzed in the same run to avoid assay variability. All markers of fibrosis were determined according to the manufacturer’s instructions. Briefly, an aliquot of the participant’s sample was added to each well following an enzyme-linked polyclonal antibody specific for human PIIINP, collagen Type IV, MMP-9 and TIMP-1. The calibrations on each microtiter plate included recombinant standards. The intensity of color developed in each well was measured using a microplate reader (Molecular Devices, Sunnyvale, CA, USA) at an absorbance of 450 nm. Samples were measured in triplicates and the mean was calculated for data analysis.

Statistical analysis was performed using the Statistica 13.3 package (TIBCO Software Inc., Cracow, Poland). Data were expressed as median, maximum and minimum values. Comparisons between study and control groups were made using Chi-2 test or the Mann–Whitney two-sample test for nonparametric data, as appropriate. The relationships between biochemical profiles were calculated by the Spearman rank-correlation test for nonparametric data. Receiver operating characteristic (ROC) curves were generated using alcohol intoxication as a classification variable and PIIINP, collagen IV, MMP-9 and TIMP-1 as prognostic variables. A statistical significance was noticed with *p* value < 0.05. The study was approved by the Bioethics Committee of the Medical University of Bialystok prior to patient recruitment, and the study is in accordance with the Declaration of Helsinki (approval number: R-I-002/337/2016, APK.002.143.2022).

## 3. Results

There were no significant differences regarding age and gender between study and control groups. We found that adolescents with alcohol intoxication had higher concentrations of MMP-9 (*p* = 0.02) and TIMP-1 (*p* = 0.007) (Figure 1 and Figure 2). No differences were observed in levels of ALT, AST, PIIINP and collagen type IV. The demographic and clinical characteristics of the analyzed groups are listed in Table 1.

Among the subjects in the study group divided by age, a higher BAC was observed in the older group (12–15 years vs. 16–17 years). TIMP-1 was the only indicator of fibrosis that differentiated both groups (Table 2). Liver parameters and fibrosis markers were similar in the control group, regardless of the alcohol concentration. (Table 3).

A correlation analysis was conducted. MMP-9 was significantly positively correlated with ALT (r = 0.38; *p* = 0.004) and total bilirubin (r = 0.39; *p* = 0.004). Positive significant correlations were also found between TIMP-1 and ALT (r = 0.47; *p* < 0.001), AST (r = 0.29; *p* = 0.03), total bilirubin (r = 0.32; *p* = 0.02) (Table 4).

ROC analysis showed that the area under the curve (AUC) of serum MMP-9 levels (cut-off = 61.0 ng/mL, Se = 71.9%, Sp = 60.9%) to predict alcohol intoxication was 0.67 (95% CI = 0.53–0.81; *p* = 0.02) (Figure 3) and for serum TIMP-1 levels (cut-off = 123.8 ng/mL, Se = 70.2%, Sp = 65.2%) was 0.69 (95% CI = 0.56–0.82; *p* = 0.003) (Figure 4). PIIINP and collagen type IV were not useful alcohol intoxication markers.

## 4. Discussion

The current research is the first report analyzing the relationship between serum levels of chosen fibrosis markers and alcohol intoxication in adolescents. In our study, we observed that even a single alcohol use significantly increased the concentration of selected liver fibrosis markers. According to our observation, markers of fibrogenesis and fibrinolysis (MMP-9 and TIMP-1) are significantly elevated in participants with alcohol intoxication, despite normal ALT, AST and total bilirubin values. In an adult study, the incidence of abnormal ALT and AST levels increased significantly from over two drinks a day. However, in multivariate analyses controlling for potential confounders, liver enzymes were not associated with significant alcohol consumption [17]. The amount of alcohol consumed by our patients was not included in the analysis due to the lack of data. Therefore, the amount of alcohol consumed by adolescents in our study cannot be compared with participants in other studies. It cannot be excluded that they consumed less alcohol than those which may affect the activity of liver enzymes.

Data showed that acute alcohol intoxication was associated with increased cytokine production, elevated oxidative stress and liver apoptosis [18]. Recent research suggests that a single episode of binge drinking may result in an imbalance in the fibrosis process [14]. MMP-9 and TIMP-1 are expressed not only in the liver but also in other organs [19]. However, due to the direct effect of ethanol on the liver (described in the introduction), it appears that the increased levels of MMP-9 and TIMP-1 may be related to hepatic metabolism. To the best of our knowledge, no studies have been published in which the damage to other organs and their effect on the concentration of fibrosis markers were simultaneously investigated.

Alcohol is one of the most commonly used addictive substance. Aiken et al. noted the average age of alcohol initiation was 15 years. Alcohol consumption in the pediatric population may not only result in physical conditions, but also increase the risk of addiction [20]. The National Survey on Drug Use and Health found ethanol use by more than 20% of adolescents between the ages of 12 and 17 [21]. According to other data, alcohol consumption among adolescents may concern more than 40% of them [22]. Hagström et al., after almost 40 years of follow-up, observed that alcohol use in young men is associated with an increased risk of severe liver disease in later life [23]. In another population-based cohort study of young men, it was observed that the risk of alcohol consumption in developing liver disease was associated with drinking to relieve a hangover [24]. The studies cited above indicate that starting alcohol use in adolescence increases the risk of liver damage. However, based on these studies, it is difficult to determine whether a single episode of alcohol consumption by adolescents could contribute to the development of liver disease. Recent studies in adults suggest that a single episode of drinking triggers excessive fibrogenesis in the liver [14].

There is increasing evidence that MMP-9 plays an important role in liver inflammation and fibrosis, due to its influence on ECM-related pathways [25]. In an animal study, mice without expression of MMP-9 had attenuated acute liver injury with the decreased level of collagen [26]. Significant differences in MMP-9 expression were also observed in studies conducted on patients with other liver diseases. In chronic hepatitis B, a significantly higher expression of MMP-9 was observed with an increase in the degree of liver fibrosis [27]. Similarly, MMP-9 has been found to be significantly elevated during hepatic fibrogenesis in patients with non-alcoholic steatohepatitis (NASH) and chronic hepatitis B [28,29]. Moreover, in patients with higher levels of MMP-9, hepatic encephalopathy was more frequently observed in the course of chronic liver disease [30]. In pediatric patients, the expression of MMP-9 was evaluated in inflammatory diseases of the gastrointestinal tract [31].

Liver fibrosis induction after alcohol consumption is associated with protein adducts produced by ethanol-derived acetaldehyde and aldehydes during lipid peroxidation. Acetaldehyde stimulates hepatic stellate cells (HSCs) to express type I collagen genes, inducing oxidative stress and activating the transforming growth factor beta (TGF) pathway, which is important in liver fibrogenesis. Alcohol also, through its negative influence on the intestinal barrier, leads to an increased amount of lipopolysaccharide (LPS) in the secondary circulation. LPS, produced by Gram-negative bacteria, increases the susceptibility of HSCs to acetaldehyde and TGF [2]. In our study we observed higher levels of MMP-9 in healthy adolescents during alcohol intoxication which may indicate that even a single episode of consumption of ethanol may have a profibrogenic effect on the liver. Prystupa et al. found increased levels of MMP-9 in alcoholics with stage C cirrhosis [32].

TIMP-1 regulates remodeling of the ECM. During liver damage, HSCs differentiate into a fibroblast-like phenotype with increased TIMP-1 expression [33]. TIMP-1 has been demonstrated to be highly related to the fibrosis stage in hepatitis C virus (HCV) mono-infected and human immunodeficiency virus (HIV)/HCV co-infected patients [34]. The assessment of TIMP-1 and other direct markers of fibrosis in serum, which has been validated as the enhanced liver fibrosis (ELF) test, is used in patients with non-alcoholic fatty liver disease (NAFLD). Its prognostic values have been reported to be sensitive, specific and reproducible in the assessment of liver fibrosis [35]. Acetaldehyde, a product of ethanol metabolism, stimulates Kupffer cells to produce reactive oxygen species (ROS). The ROS activate intracellular pro-fibrogenic pathways in HSCs, including the TIMP-1 pathways [2]. In our study, we found that the serum concentration of TIMP-1 in adolescents admitted to the hospital due to alcohol intoxication was significantly high. This may indicate that even a single binge drinking episode may contribute to the development of liver disease in adolescents. Alcohol-induced liver damage was also seen in another study. Acute alcohol intoxication caused rapid changes in circulating lipids. Free fatty acids and lysophosphatidylcholine, potentially lead to lipoapoptosis and steatohepatitis [36].

Collagen type IV is a key component of the basement membrane, one of the compartments of ECM. In an immunohistochemical assessment, collagen type IV was detected in the early stage of alcohol-related liver disease and was significantly elevated with a higher fibrosis stage [37]. Moreover, elevated levels of type IV collagen in serum are associated with the degree of fibrosis and the severity of hepatitis [38]. According to our observation, this biomarker of fibrosis did not differ between study and control groups.

Markers of type III collagen formation, such as PIIINP and procollagen III (PRO-C3), are promising molecules for the detection and monitoring of the fibrosis stage in patients with chronic liver diseases [38,39]. We found no significant differences in the concentration of PIIINP between healthy controls and patients with alcohol intoxication. To date, one study has been published comparing the effects of binge drinking on ECM formation in patients with alcohol-related liver disease, NAFLD and healthy controls. Measurements of markers of collagen formation and degradation showed a significant increase in PRO-C3 in all participants without significant differences between groups. However, it is difficult to compare our research due to the different methodology. Torp et al. measured the biomarkers at baseline and after consuming the planned dose of alcohol in hepatic and systemic circulation [14]. Moreover, the degree of puberty in children may influence the concentration of markers of type III collagen synthesis [40]

To the best of our knowledge, we analyzed biomarkers of liver fibrosis in adolescents with alcohol intoxication for the first time. However, some limitations of the study should be mentioned. The low number of enrolled patients was the main limitation of our work. Another limitation was the lack of monitoring of patients over an extended period of time. In addition, we only analyzed biomarker levels in the systemic circulation due to the invasive nature of measuring a blood sample from the hepatic vein. Our study only provides preliminary information on MMP-9 and TIMP-1 as potential biomarkers of alcohol intoxication, which should now be investigated in a multicenter study.

## 5. Conclusions

Our results show that even a single episode of alcohol intoxication in adolescents can lead to an imbalance in markers of fibrosis. Moreover, MMP-9 and TIMP-1 appear to be more sensitive markers of hepatocyte damage than ALT and AST. Further multi-center studies involving a larger group of patients are needed to be able to more accurately assess the effect of alcohol consumption on hepatic fibrogenesis and fibrolysis.

## Figures and Tables

**Figure 1 biomolecules-12-00710-f001:**
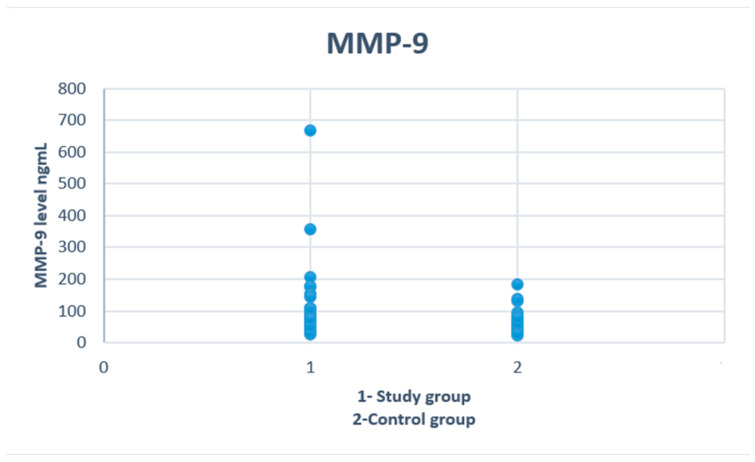
Serum concentrations of MMP-9 in study and control groups.

**Figure 2 biomolecules-12-00710-f002:**
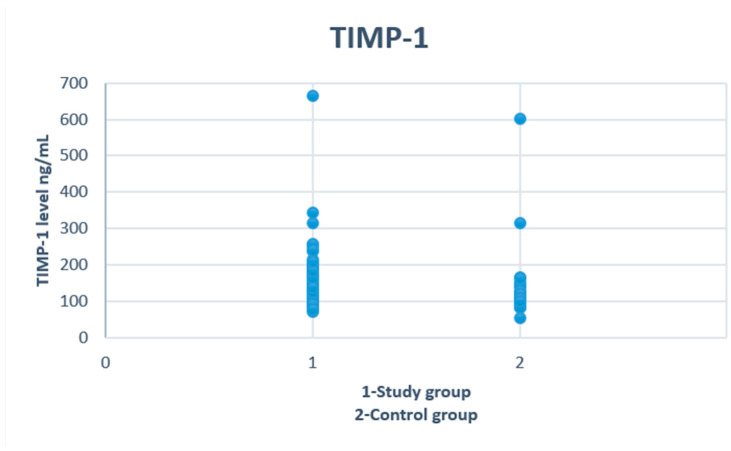
Serum concentrations of TIMP-1 in study and control groups.

**Figure 3 biomolecules-12-00710-f003:**
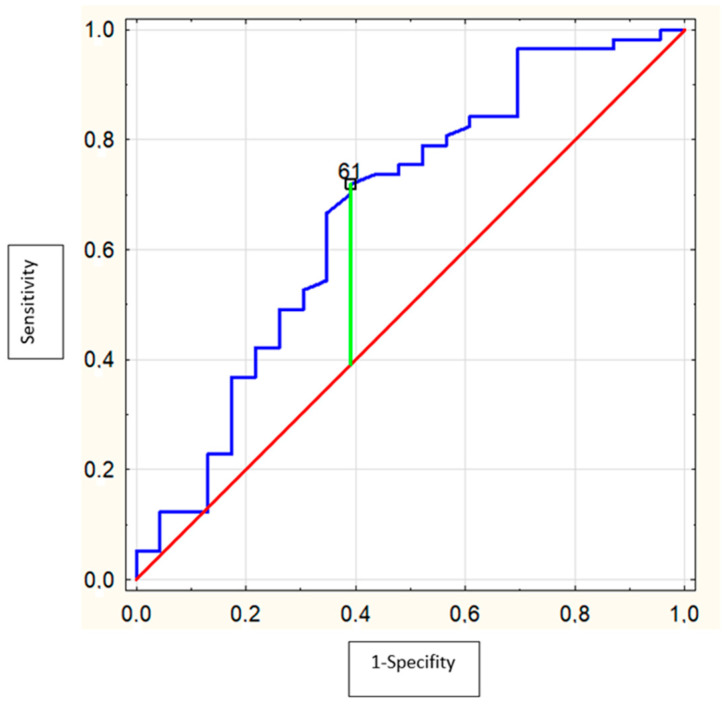
Receiver operating characteristics (ROC) curve of the ability of MMP-9 to differentiate the adolescents with alcohol intoxication (AUC = 0.67 (95%confidence interval: 0.53–0.81; *p* = 0.02) with optimal cut-off value of 61.0 ng/mL).

**Figure 4 biomolecules-12-00710-f004:**
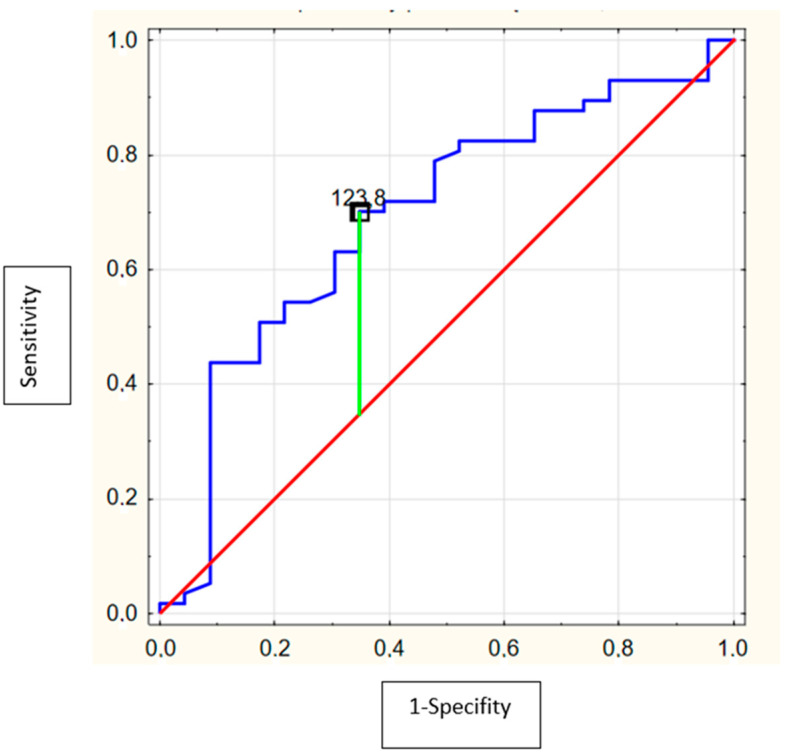
Receiver operating characteristics (ROC) curve of the ability of TIMP-1 to differentiate the adolescents with alcohol intoxication. (AUC = 0.69 (95%confidence interval: 0.56–0.82; *p* = 0.003) with optimal cut-off value of 123.8 ng/mL).

**Table 1 biomolecules-12-00710-t001:** Characteristics of the study and control groups.

Parameter	Study Group (n = 57)	Control Group (n = 24)	*p*
Alcohol (g/L)	1.78 (0.48–3.26)	-	NA
Age (years)	15 (12–17)	16 (12–17)	NS
ALT (IU/L)	13 (7–48)	13 (11–15)	NS
AST (IU/L)	22 (14–45)	20 (14–29)	NS
PIIINP (ng/mL)	25.66 (14.34–48.42)	24.67 (12.96–40.82)	NS
Collagen IV (ng/mL)	68.11 (40.72–121.1)	70.79 (43.16–108.1)	NS
MMP-9 (ng/mL)	72.4 (26.4–667.4)	58 (22.8–183.0)	0.02 *
TIMP-1 (ng/mL)	153.4 (71.1–665.0)	110.3 (54.0–601.0)	0.007 *
MMP-9/TIMP-1 ratio	0.5 (0.21–1.04)	0.52 (0.22–1.15)	NS

Alanine aminotransferase (ALT), aspartate aminotransferase (AST), amino terminal propeptide of type III procollagen (PIIINP), matrix metallopeptidase 9 (MMP-9) and tissue inhibitor of metalloproteinase 1 (TIMP-1), not significant (NS), not applicable (NA) (* *p* < 0.05).

**Table 2 biomolecules-12-00710-t002:** Characteristics of the study group in terms of age.

Parameter	12–15 Years (n = 30)	16–17 Years (n = 27)	*p*
Alcohol (g/L)	1.64 (0.48–3.26)	2.06 (0.51–3.06)	0.02 *
ALT (IU/L)	13 (9–48)	13 (7–38)	NS
AST (IU/L)	21 (14–45)	22 (15–35)	NS
Bilirubin (mg/dL)	0.35 (0.15–1.49)	0.32 (0.15–1.57)	NS
CK (IU/L)	218 (88–923)	193 (79–624)	NS
CK-MB (IU/L)	21 (13–32)	19 (11–55)	NS
Urea (mg/dL)	23 (12–35)	21 (10–36)	NS
PIIINP (ng/mL)	27.74 (14,34–45.9)	24.81 (19–48.42)	NS
Collagen IV (ng/mL)	73.59 (40.72–121.1)	67.59 (50.94–120.29)	NS
MMP-9 (ng/mL)	69.2 (26.4–667.4)	74.6 (39.8–355.4)	NS
TIMP-1 (ng/mL)	134.55 (71.1–665)	176.1 (96.6–342.8)	0.03 *
MMP-9/TIMP-1 ratio	0.57 (0.21–1)	0.43 (0.21–1.04)	NS

Alanine aminotransferase (ALT), aspartate aminotransferase (AST), creatine kinase (CK), creatine kinase-MB (CK-MB), amino terminal propeptide of type III procollagen (PIIINP), matrix metallopeptidase 9 (MMP-9) and tissue inhibitor of metalloproteinase 1 (TIMP-1), not significant (NS), (* *p* < 0.05).

**Table 3 biomolecules-12-00710-t003:** Characteristics of the study group in terms of BAC.

Parameter	BAC < 1.78 (n = 28)	BAC ≥ 1.78 (n = 29)	*p*
Age (year)	15 (12–17)	16 (12–17)	NS
ALT (IU/L)	13 (7–38)	13 (8–48)	NS
AST (IU/L)	22 (15–45)	22 (14–42)	NS
Bilirubin (mg/dL)	0.31 (0.15–1.49)	0.33 (0.15–1.57)	NS
CK (IU/L)	218 (88–923)	202 (79–624)	NS
CK-MB (IU/L)	20 (11–43)	20 (13–55)	NS
Urea (mg/dL)	23 (12–35)	21 (10–36)	NS
PIIINP (ng/mL)	25.66 (14.34–40.18)	26.05 (18.3–48.42)	NS
Collagen IV (ng/mL)	67.45 (40.72–104.32)	69.77 (46.34–121.1)	NS
MMP-9 (ng/mL)	69.2 (26.4–667.4)	74.6 (39.8–205.2)	NS
TIMP-1 (ng/mL)	148.8 (71.7–665)	163.3 (83.6–253.8)	NS
MMP-9/TIMP-1 ratio	0.59 (0.21–1)	0.43 (0.21–1.04)	NS

Alanine aminotransferase (ALT), aspartate aminotransferase (AST), creatine kinase (CK), creatine kinase-MB (CK-MB), amino terminal propeptide of type III procollagen (PIIINP), matrix metallopeptidase 9 (MMP-9) and tissue inhibitor of metalloproteinase 1 (TIMP-1), not significant (NS).

**Table 4 biomolecules-12-00710-t004:** Correlations of fibrosis markers with laboratory results in study group.

	PIIINP	Collagen IV	MMP-9	TIMP-1	MMP-9/TIMP-1 Ratio
ALT	R = 0.28 *p* = 0.04 *	R = 0.34 *p* = 0.008 *	R = 0.38 *p* = 0.004 *	R = 0.47 *p* < 0.0001 *	R = 0.3; *p* = 0.02 *
AST	R = 0.42 *p* = 0.001 *	R = 0.42 *p* = 0.001 *	R = 0.23 *p* = 0.09 *	R = 0.29 *p* = 0.03 *	R = 0.3; *p* = 0.01 *
Total bilirubin	R = 0.44 *p* = 0.001 *	R = 0.44 *p* = 0.001 *	R = 0.39 *p* = 0.004 *	R = 0.32 *p* = 0.018 *	NS
CK	R = 0.3 *p* = 0.03 *	R = 0.3 *p* = 0.03 *	R = 0.24 *p* = 0.08 *	R = 0.28 *p* = 0.04 *	R = 0.38; *p* = 0.005 *
CK-MB	R = 0.33 *p* = 0.01 *	NS	NS	NS	R = 0.3; *p* = 0.03 *
Urea	R = 0.35 *p* = 0.008 *	R = 0.35 *p* = 0.008 *	NS	NS	R = 0.28; *p* = 0.04 *

Alanine aminotransferase (ALT), aspartate aminotransferase (AST), creatine kinase (CK), creatine kinase-MB (CK-MB), amino terminal propeptide of type III procollagen (PIIINP), matrix metallopeptidase 9 (MMP-9) and tissue inhibitor of metalloproteinase 1 (TIMP-1), not significant (NS), (* *p* < 0.05).

## Data Availability

Data supporting reported results are available from the first author to all interested researchers.

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
