# Peer review of "Increase in Serum MMP-9 and TIMP-1 Concentrations during Alcohol Intoxication in Adolescents—A Preliminary Study"

_biomolecules, 2022, doi:10.3390/biom12050710_

Round 1
Reviewer 1 Report
To the authors of the manuscript,
In the present work the increased MMP-9 and TIMP-1 serum levels in alcohol intoxication patients is well characterized. However, I would recommend to perform some amendment prior acceptance of the work.
The increase of both proteins is characterized independently of the age and the BAC. These should be deeply analyzed in Tables 2 and 4.
In the tables, with Table 4 in particular, I recommend to highlight the statistically significant values with, or ***.
A further discussion about MMP-9 and TIMP-1 as predictive markers would be recomendable. The values stablished of 0.67 and 0.69 are really near to the average levels of the control group.
Author Response
Reviewer 1
Thank you for your very valuable comments. Please find our answers below.
-The increase of both proteins is characterized independently of the age and the BAC. These should be deeply analyzed in Tables 2 and 4.
Answer: The information has been completed.
-In the tables, with Table 4 in particular, I recommend to highlight the statistically significant values with, or ***.
Answer: The information has been completed.
-A further discussion about MMP-9 and TIMP-1 as predictive markers would be recomendable. The values stablished of 0.67 and 0.69 are really near to the average levels of the control group.
Answer: The aim of our research was not to identify new predictive markers of alcohol intoxication, but to answer whether we observe changes in the concentration of fibrosis markers in children with even single alcohol use. Without invasive diagnostic, such as liver biopsy (no indications for liver biopsy), it is difficult to determine whether the increase in MMP-9 and TIMP-1 concentrations is equivalent to an increase in fibrosis in the histopathological examination. Therefore, more research is needed. Despite the relatively low AUC values, they were statistically significant.
Reviewer 2 Report
this is an interesting study.
My two main comments are:
1) do the authors know for sure that this is 'single alcohol use' or are they assuming this? How do they know that these individuals are not chronic drinkers?
2) Have other diseases esp NAFLD and viral hep being excluded?
Some minor comments:
1) I am not sure that the correlation of low levels of AST/ALT (normal) are relevant in this manuscript
2) I am not sure of the importance of higher blood alcohol levels in 12-15 vs 16-17 age groups.
Author Response
Thank you for your very valuable comments. Please find our answers below.
-do the authors know for sure that this is 'single alcohol use' or are they assuming this? How do they know that these individuals are not chronic drinkers?
-Have other diseases esp NAFLD and viral hep being excluded?
Answer: The information was obtained from the medical history and laboratory test results
-I am not sure that the correlation of low levels of AST/ALT (normal) are relevant in this manuscript
Answer: Due to the observed statistical significance, the data remained in the manuscript. Depending on the decision, the information may be transferred to "Supplementary materials".
-I am not sure of the importance of higher blood alcohol levels in 12-15 vs 16-17 age groups.
Answer: Due to the observed statistical significance, the data remained in the manuscript. Depending on the decision, the information may be transferred to "Supplementary materials".
Reviewer 3 Report
The manuscript entitled “Increase in serum MMP-9 and TIMP-1 concentrations during alcohol intoxication in adolescents” suggested non-invasive biomarkers of liver fibrosis during alcohol intoxication. The authors conducted prospective study including 57 adolescents aged 14-17 years admitted to the ER from 2017-2018. This article is very straightforward and the explanation is interesting to provide a new insight to readers about new biomarkers. However, there are several concerns must be addressed.
<concerns>
- The author of this article claimed that MMP-9 and TIMP-1 play an important role in non-invasive biomarkers of liver fibrosis. However, those samples were collected from mainly young people after binge alcohol drinking. Excessive alcohol consumption (single episode of drinking) is highly associated with immune cell infiltration in liver, ER stress and hepatocyte death, not the progression of liver fibrosis. Therefore, the conclusion (MMP-9 and TIMP-1 as bio-markers in liver fibrosis) from the authors finding was not too much related with alcohol intoxication.
- The authors demonstrated that MMP-9 and TIMP-1 are efficient liver injury biomarkers, however, the authors did not compare the expression of MMP- and TIMP-1 in adult population (aged over 17). Without the data from adult, it is hard to accept MMP-9 and TIMP-1 as non-invasive biomarkers of alcohol intoxication in adolescents.
- The authors also claimed that MMP-9 and TIMP-1 to be more sensitive markers of hepatocyte damage than ALT and AST. However, MMP-9 and TIMP-1 are not mainly expressed in liver (https://www.ncbi.nlm.nih.gov/gene/4318) (https://www.ncbi.nlm.nih.gov/gene/7076). (in contrast, ALT, AST are highly expressed in liver). Must be considered off-target effects, please explain more about this concern.
- The authors evaluated serum MMP-9 and TIMP-1 using ELISA. In general, ELISA is time-consuming process to obtain the results (several hours to overnight) compared to the evaluation of ALT and AST. Please explain more about this concern.
- In table2, the TIMP-1 data is missing.
- In table3, TIMP-1 and MMP-9 data are missing.
- In Table1, only MMP-9 and TIMP-1 have statistical significance, however, it is difficult to understand at-a-glance. Put “dot graph” (not bar graph) would be helpful for better understanding.
- In “Materials and Methods”, add some more details how to evaluate blood chemistry analyses (instrument model or cat number…)
- In the part of “Introduction” (page2), TIPM-1 à TIMP-1
- In the part of “Results (page3), 12-5 years vs 16-17 years à 12-15 years vs 16-17 years
Author Response
Thank you for your very valuable comments. Please find our answers below.
-The author of this article claimed that MMP-9 and TIMP-1 play an important role in non-invasive biomarkers of liver fibrosis. However, those samples were collected from mainly young people after binge alcohol drinking. Excessive alcohol consumption (single episode of drinking) is highly associated with immune cell infiltration in liver, ER stress and hepatocyte death, not the progression of liver fibrosis. Therefore, the conclusion (MMP-9 and TIMP-1 as bio-markers in liver fibrosis) from the authors finding was not too much related with alcohol intoxication.
Answer: LINE 184-186: “Data showed that acute alcohol intoxication was associated with increased cytokine production, elevated oxidative stress and liver apoptosis [18]. Recent research suggests that even a single episode of binge drinking may result in an imbalance in the fibrosis process [12].” In accordance with the reviewer's suggestion, the conclusions were changed.
-The authors demonstrated that MMP-9 and TIMP-1 are efficient liver injury biomarkers, however, the authors did not compare the expression of MMP- and TIMP-1 in adult population (aged over 17). Without the data from adult, it is hard to accept MMP-9 and TIMP-1 as non-invasive biomarkers of alcohol intoxication in adolescents.
Answer: Adult studies were conducted by Torp et. al, in which an imbalance in markers of fibrosis was demonstrated in patients with alcohol intoxication (this study inspired our analysis). We deliberately examined pediatric patients in our study to see if there are similar relationships. In addition, due to the fact that the study was performed in a pediatric hospital, it was impossible to include adult patients.
-The authors also claimed that MMP-9 and TIMP-1 to be more sensitive markers of hepatocyte damage than ALT and AST. However, MMP-9 and TIMP-1 are not mainly expressed in liver (https://www.ncbi.nlm.nih.gov/gene/4318) (https://www.ncbi.nlm.nih.gov/gene/7076). (in contrast, ALT, AST are highly expressed in liver). Must be considered off-target effects, please explain more about this concern.
Answer: LINE: 186-191 “MMP-9 and TIMP-1 are expressed not only in the liver but also in other organs [c]. However, due to the direct effect of ethanol on the liver (described in the introduction), it appears that the increased levels of MMP-9 and TIMP-1 may be related to hepatic metabolism. To the best of our knowledge, no studies have been published in which the damage to other organs and their effect on the concentration of fibrosis markers were simultaneously investigated”.
Moreover, the elevation of ALT and AST is observed not only in the liver diseases (https://www.ncbi.nlm.nih.gov/books/NBK425/).
-The authors evaluated serum MMP-9 and TIMP-1 using ELISA. In general, ELISA is time-consuming process to obtain the results (several hours to overnight) compared to the evaluation of ALT and AST. Please explain more about this concern.
Answer: We agree with the reviewer that the determination of ALT and AST is performed by faster methods. Information on how to perform determinations has been added in "Materials and methods".
- In table2, the TIMP-1 data is missing.
Answer: The information has been completed.
- In table3, TIMP-1 and MMP-9 data are missing.
Answer: The information has been completed.
- In Table1, only MMP-9 and TIMP-1 have statistical significance, however, it is difficult to understand at-a-glance. Put “dot graph” (not bar graph) would be helpful for better understanding.
Answer: The graphs have been added.
- In “Materials and Methods”, add some more details how to evaluate blood chemistry analyses (instrument model or cat number…)
Answer: LINE 82-85 “Routine biochemical analyses including alanine aminotransferase (ALT), and addition-ally in study group also total bilirubin, creatine kinase (CK), creatine kinase-MB (CK-MB) and urea were measured on the day of blood collection by the standard clinical laboratory techniques, using an automatic analyzer Dimension AR (Dade Behring, Newark, New Jersey, United States).”
Reviewer 4 Report
Increase in serum MMP-9 and TIMP-1 concentrations during 2 alcohol intoxication in adolescents - a preliminary study.........
.
...
Katarzyna Zdanowicz 1*, Monika Kowalczuk-Kryston 1 , Witold Olanski,Irena Werpachowska 1 , Wlodzimierz Mielech2 , Dariusz Marek Lebensztejn
1 ,My comments and suggestions for authors are as follows: for understanding.
1. Alcohol intoxication and early alcohol effect of causing fibrogenesis should have been described with commentes given some supprting citation . How could disease develop in a short time.
2 Well known makers of fibrosis are PIIINP and TIPM-1 and markers fibrnolysis are MMPS should be discussed in more details.
3 Material and Method Section .Line 81.
Blood alcohol Coc were determined using head space gas chromatograhy ,Please mention referenceor methodology uesd.
4. Some explainaton needed as to how fibrogenesis and Fibrinolysis(MMP-9and TMP- 1)are significanly elivated in partcipant with alcohol in toxication despite ALT ,AST Total bilirubin normal valuse(Ref line148-151) .This need some explaination.
5.Line 221.The authors mentioned some limitatom of study.So what are those ,shold be mentioned.
6.Line 82- 83 .Material and Method.Serum PIIINP, Collagen Type IV,MMP-9 and TIMP-1 levels were measured using a double -antibodysandwhich ELISA with laboratory kits(Cloud-Clone Corp Katy,TX,77494,USA.Please describe in brief methodology.
This study is a follow-up to work on liver damage from alcohol intoxication. The aim of this study was to explore the early effects of alcohol intoxication on liver fibrosis in adolescents.
Serum 18 levels of amino terminal propeptide of type III procollagen (PIIINP), type IV collagen, matrix met- 19 allopeptidase 9 (MMP-9) and tissue inhibitor of metalloproteinase 1 (TIMP-1) were determined by 20 enzyme-linked immunosorbent assays. Results: There were significant differences in MMP-9 21 (p=0.02) and TIMP-1 (p=0.007) levels between the study and control groups.
Conclusion: MMP-9 and TIMP-1 are promising, early, non-invasive biomarkers of liver fibrosis during alcohol intoxication in adolescents. .
.
Suggested points to include in the manuscript and discuss will improve the
manuscript.
Author Response
Thank you for your very valuable comments. Please find our answers below.
-Alcohol intoxication and early alcohol effect of causing fibrogenesis should have been described with comments given some supporting citation . How could disease develop in a short time.
Answer: Recent studies in adults suggest that a single episode of drinking triggers excessive fibrogenesis in the liver [Torp N, Israelsen M, Nielsen MJ, Åstrand CP, Juhl P, Johansen S, et al. Binge drinking induces an acute burst of markers of hepatic fibrogenesis (PRO-C3). Liver Int. 2022;42(1):92-101]. This observation proves that the concept of a "safe" amount of alcohol should be re-examined and that binge drinking can be much more severe than previously assumed.
-Well known makers of fibrosis are PIIINP and TIPM-1 and markers fibrnolysis are MMPS should be discussed in more details.
Answer: LINE 54-59 “PIIINP is released during the synthesis and deposition of type III collagen. The amount of molecules produced and PIIINP released is in a stoichiometric ratio of 1: 1, so this protein is useful in the evaluation of fibrogenesis processes [11]. MMPs are calcium-dependent zinc-containing peptidases. They are included among the useful markers of fibrinolysis due to their participation in the degradation and turnover of most components in the ECM. TIMPs, as inhibitors of MMPs activity, take part in the process of fibrosis [12].”
-Material and Method Section .Line 81.
Blood alcohol Coc were determined using head space gas chromatograhy ,Please mention referenceor methodology uesd.:
Answer: LINE 86-88 “Blood alcohol concentration (BAC) determinations were performed with headspace gas chromatography according to Cordell et al. [16].”
-Some explanation needed as to how fibrogenesis and Fibrinolysis(MMP-9and TMP- 1)are significantly elevated in participant with alcohol intoxication despite ALT ,AST Total bilirubin normal values (Ref line148-151) .This need some explanation.
Answer: LINE 174-176 “According to our observation, markers of fibrogenesis and fibrolysis (MMP-9 and TIMP-1) are significantly elevated in participants with alcohol intoxication, despite normal ALT, AST and total bilirubin values and seem to respond to a single alcohol use. In an adult study, the incidence of abnormal ALT and AST levels increased significantly from zero to over two drinks a day. However, in multivariate analyzes controlling for potential confounders, liver enzymes were not associated with significant alcohol consumption [17]. The amount of alcohol consumed by our patients was not included in the analysis due to the lack of data, therefore the amount of alcohol consumed by adolescents entering our study cannot be compared with participants in other studies, and it cannot be ruled out that they consumed less alcohol than those which may affect the activity of liver enzymes”.
-Line 221.The authors mentioned some limitation of study. So what are those ,should be mentioned
Answer: LINE 261-264 “However, some limitation of the study should be mentioned. The low number of enrolled patients was the main limitation of our work. Another limitation was the lack of monitoring of patients over an extended period of time. In addition, we only analyzed biomarker levels in the systemic circulation due to the invasive nature of measuring a blood sample from the hepatic vein. Our study only gives preliminary information about MMP-9 and TIMP-1 as a potential biomarker of alcohol intoxication, which should be investigated in further researches.”
-Line 82- 83 .Material and Method. Serum PIIINP, Collagen Type IV,MMP-9 and TIMP-1 levels were measured using a double –antibody sandwich ELISA with laboratory kits(Cloud-Clone Corp Katy,TX,77494,USA.Please describe in brief methodology.
Answer: LINE 93-99 “Collected samples from the whole participants were analyzed in the same run to avoid assay variability. All markers of fibrosis were determined according to the manufacturer’s instructions. Briefly, an aliquot of the participant’s sample was added each well following an enzyme-linked polyclonal antibody specific for human PIIINP, collagen Type IV, MMP-9 and TIMP-1. The calibrations on each microtiter plate included recombinant standards. The intensity of color developed in each well was measured using a microplate reader (Molecular Devices, Sunnyvale, CA, USA) at an absorbance of 450 nm. Samples were measured as triplicates and the mean was calculated for data analysis.”
Round 2
Reviewer 3 Report
All concerns have been addressed.